# ONLINE DOMAIN INDEXING

## ABSTRACT

Domain adaptation (DA) in real-world applications often unfolds in an online fashion, where data arrives sequentially with limited domain access and imbalanced sampling across domains. For example, in personalized ads prediction, users from different demographic groups (e.g., countries or age cohorts) correspond to distinct domains with highly skewed data availability, and user interests evolve over time. Recent work has explored domain indices to capture latent inter-domain relationships and improve adaptation (Wang et al., 2020; Xu et al., 2023). However, existing methods such as Variational Domain Indexing (VDI) (Xu et al., 2023) assume full domain observability and balanced mini-batches, limiting their applicability to real-world scenarios with online domain shift and data imbalance. To address these challenges, we propose Online Domain Indexing (ODI), the first continual domain indexing and adaptation framework designed for partial domain access and inter-domain sample imbalance. Starting from a base model pretrained on historical source and target domains, ODI incrementally updates domain indices over time using a smoothed reweighting kernel and a replay buffer to ensure stable adaptation. Experiments on both synthetic and real-world datasets demonstrate that ODI consistently outperforms state-of-the-art baselines in long-term accuracy under dynamic and resource-constrained conditions.

## 1 INTRODUCTION

Modern machine learning systems often operate in dynamic environments, where data distributions evolve over time and models must continually adapt to incoming data streams. This challenge is amplified in domain adaptation (DA) settings, where data is collected from multiple source and target domains, each associated with its own distribution. In practice, data from different domains often does not arrive simultaneously; instead, it comes in sequential rounds, often with partial domain coverage (e.g., Domains $1 \sim 3$ in Round 1 and Domains $4 \sim 6$ in Round 2) and imbalanced sample sizes across domains.

This setting naturally arises in real-world applications such as personalized advertising. For example, on platforms like Facebook, users from different countries or age groups often exhibit distinct click behaviors and preferences; these users naturally correspond to separate domains. These domains not only exhibit heterogeneous behavior, but also vary significantly in population size, often leading to domain-level imbalance (i.e., variation in the number of data points across domains). Models are typically trained to predict whether a user will click on an ad based on user and item embeddings. However, as user interests evolve over time, the underlying domain distributions shift. This necessitates an online domain adaptation framework capable of handling domain heterogeneity, data imbalance, and temporal drift.

While prior work in domain adaptation has focused extensively on static settings with full domain observability, recent methods have also explored learning domain indices to capture latent inter-domain relationships and improve adaptation (Wang et al., 2020; Xu et al., 2022c; 2023). In particular, Variational Domain Indexing (VDI) (Xu et al., 2023) has shown promise in inferring domain indices while improving accuracy. However, these approaches typically assume balanced mini-batches and full access to all domains during training, limiting their applicability to online scenarios with distributional shift, partial observability, and data sparsity.

In this paper, we study a realistic and underexplored setting called Online Imbalanced Domain Adaptation (OIDA). We assume access to a fixed set of $C$ domains, including labeled source domains

and unlabeled target domains. In each round, a new dataset arrives, representing a slight shift from the historical distribution over these $C$ domains. However, the model is only able to access data from a randomly selected subset of $k \ll C$ domains, and the number of available data points for each domain is imbalanced; some domains may provide significantly more data than others. This introduces dual challenges: *limited domain observability* and *cross-domain data imbalance*.

To address these challenges, we propose Online Domain Indexing (ODI), the first general framework to extend VDI into the online regime under realistic access constraints. Our method begins by initializing a base VDI model using historical data from all domains. Then, in each round, the model is incrementally updated using the new, partially accessible and imbalanced data. It predicts domain-specific representations, i.e., domain indices, and labels while continually adapting to distribution shifts across rounds.

Our key contributions are as follows:

- We formulate the Online Imbalanced Domain Adaptation (OIDA) problem, where in each round, the learner receives data from only $k \ll M$ domains, and the number of data points available for each domain is imbalanced.
- We propose Online Domain Indexing (ODI), a general learning framework that incrementally updates domain indices under domain access constraints and sample imbalance.
- Extensive experiments on both synthetic and real-world datasets demonstrate that ODI significantly outperforms existing domain adaptation baselines in long-term prediction accuracy under distribution shift and data sparsity.

## 2 RELATED WORK

**Typical Domain Adaptation.** Domain adaptation (DA) has been extensively studied (Wang et al., 2020; Xu et al., 2023; Pan & Yang, 2009; Pan et al., 2010; Long et al., 2018; Saito et al., 2018; Sankaranarayanan et al., 2018; Zhang et al., 2019; Peng et al., 2019; Chen et al., 2019; Dai et al., 2019; Nguyen-Meidine et al., 2021), with the primary goal of aligning distributions between labeled source and unlabeled target domains to enable effective generalization. This alignment is typically achieved either through direct matching of distributional statistics (Pan et al., 2010; Tzeng et al., 2014; Sun & Saenko, 2016; Peng et al., 2019; Nguyen-Meidine et al., 2021) or via adversarial training (Ganin et al., 2016; Zhao et al., 2017; Tzeng et al., 2017; Zhang et al., 2019; Kuroki et al., 2019; Chen et al., 2019; Dai et al., 2019; Wang et al., 2020; Xu et al., 2022b; Liu et al., 2023; Shi & Wang, 2023). The latter has gained popularity due to its strong theoretical foundations (Goodfellow et al., 2014; Zhao et al., 2018; 2019), compatibility with deep architectures, and empirical success.

**Domain-Index-Based Domain Adaptation.** Recent works improve DA by generating domain identities from data and performing multi-domain learning. These domain identities can be ordinal (Deecke et al., 2021; Du et al., 2021; Lu et al., 2022) or continuous (Wang et al., 2020; Xu et al., 2022c; 2023). In classification tasks, Peng et al. (2020) embed visual domains into vectors to represent inter-domain similarities. Xu et al. (2023) formalizes the notion of a "domain index" and proposes a generative model to infer it. Unlike prior works that generate domain indices explicitly, VDI (Xu et al., 2023) treats them as latent variables and infers them using variational inference algorithms. However, all these methods assume an offline setting with full domain observability and balanced samples. In contrast, our approach addresses online adaptation under partial access and long-tailed domain distributions – a substantially harder and more realistic setting.

**Gradual and Online Domain Adaptation.** Our work also relates to gradual DA, which introduces intermediate unlabeled domains to bridge the gap between a fixed source domain and a fixed target domain (Bobu et al., 2018; Chen & Chao, 2021; He et al., 2024; Sagawa & Hino, 2022; Zhuang et al., 2024; Najafi et al., 2024). However, these methods assume static source and target domains. In contrast, our setting involves temporally evolving source and target domains across training rounds.

Different from gradual DA, online DA goes beyond a fixed target domain and addresses continuous domain shifts without clear domain boundaries. For example, Volpi et al. (2022) propose an online DA evaluation protocol; Panagiotakopoulos et al. (2022) design a static-dynamic teacher coordination framework; Colomer et al. (2023) develop a hardware-aware real-time adaptation scheme; Liu et al. (2024) detect and adapt to domain shifts in real-time under uncertainty.

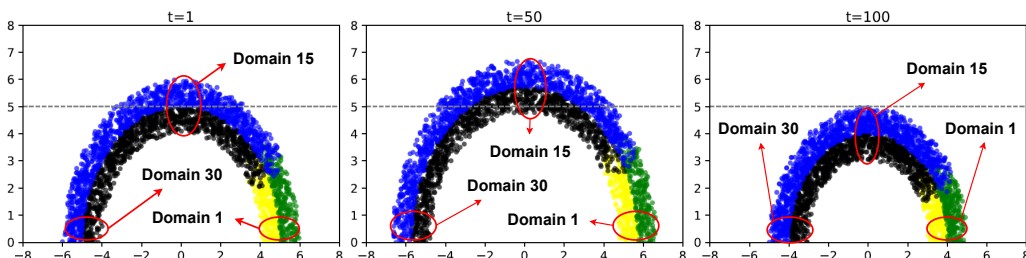

Figure 1: The Online Circle dataset with 30 evolving domains, including 6 source domains and 24 target domains. The data distribution gradually shifted from Round $t = 1$ to Round $t = 100$.

However, both gradual and online DA methods assume a *single* evolving domain. They are therefore not applicable to our setting where there are *multiple* evolving domains, including multiple evolving source and target domains. Fig. 1 shows an example dataset in our setting, with 6 evolving source domains (marked as yellow and green) and 24 evolving target domains (marked as black and blue) across 100 rounds.

## 3 METHODS

**Notations.** We use $[n]$ to denote the set $\{1, 2, \cdots, n\}$. A tuple $(\mathbf{x}, y, k)$ denotes a single data sample, where $\mathbf{x} \in \mathbb{R}^d$ is the input feature vector, $y \in \mathbb{R}$ is the label, and $k \in [K]$ indicates the domain identity. We denote the vector $\mathbf{x}$'s Euclidean norm as $\|\mathbf{x}\|_2 = \sqrt{\mathbf{x}^\top \mathbf{x}}$. For a matrix $\mathbf{W} \in \mathbb{R}^{m \times n}$, $\|\mathbf{W}\|$ denotes its operator norm and $\|\mathbf{W}\|_F$ its Frobenius norm. We denote as $\mathbf{z} \in \mathbb{R}^{d_z}$ the data encoding generated from an encoder that takes x as input. We use $I(\cdot; \cdot)$ to denote mutual information.

**Problem Setting.** We consider an *online unsupervised domain adaptation* setting where data arrives sequentially over a series of rounds $t = 1, 2, \cdots, T$. Across all rounds, the full set of domains is denoted as $[K]$, partitioned into labeled *source domains* $\mathcal{K}_S \subset [K]$ and unlabeled *target domains* $\mathcal{K}_T \subset [K]$, where $\mathcal{K}_S \cup \mathcal{K}_T = [K]$. Each domain $k \in [K]$ contains $D_k$ samples. We denote a source-domain sample as $(\mathbf{x}, y, k)$ and a target-domain sample as $(\mathbf{x}, k)$, where labels $y$ are unavailable.

We begin with access to a historical dataset $\mathcal{D}_{hist}$ consisting of balanced samples from all domains, which is used to pre-train a domain adaptation model under full domain observability. During online adaptation stage, in each round $t \in [T]$, we receive an evolved dataset $\mathcal{D}^{(t)}$ containing only a *small subset* of source and target domains $\mathcal{K}_S^{(t)} \subset \mathcal{K}_S$ and $\mathcal{K}_T^{(t)} \subset \mathcal{K}_T$, with imbalanced data availability across domains. The goal is to predict the labels of all target domains across *all rounds* accurately.

### 3.1 PRELIMINARY: VARIATIONAL DOMAIN INDEXING (VDI)

We build upon the Variational Domain Indexing (VDI) framework (Xu et al., 2023), which provides a principled way to represent domain identity as continuous latent vectors. Intuitively, one can think of each domain as being assigned a *low-dimensional* embedding (e.g., 2 dimensions) – the *domain index* – that captures its position relative to other domains. Domains with similar characteristics are mapped to nearby embeddings, while those with very different properties are placed farther apart. For instance, when modeling temperature across the 50 U.S. states, California and Nevada would have domain indices close to each other due to similar climates, whereas California and Massachusetts would lie farther apart. This continuous view of domain identity allows us to measure similarity between domains and to smoothly adapt models across heterogeneous environments.

Formally, VDI assigns each domain $k$ a global domain index vector $\boldsymbol{\beta}_k \in \mathbb{R}^{d_\beta}$, and each sample $i$ a local domain index $\mathbf{u}_i \in \mathbb{R}^{d_u}$. These indices serve as latent coordinates that complement the data encoding $\mathbf{z}$, enabling the model to disentangle domain-specific variations from label-relevant features. Compared to discrete domain labels, continuous indices provide two advantages: (i) they capture graded relationships among domains rather than treating them as independent categories, and (ii) they offer a compact representation that scales naturally when the number of domains grows.

**Generative Objective and Graphical Model.** VDI is trained using a variational objective that integrates generative modeling with adversarial regularization. Let $\boldsymbol{\alpha} = \{\mu_\alpha, \sigma_\alpha\}$ be the prior for

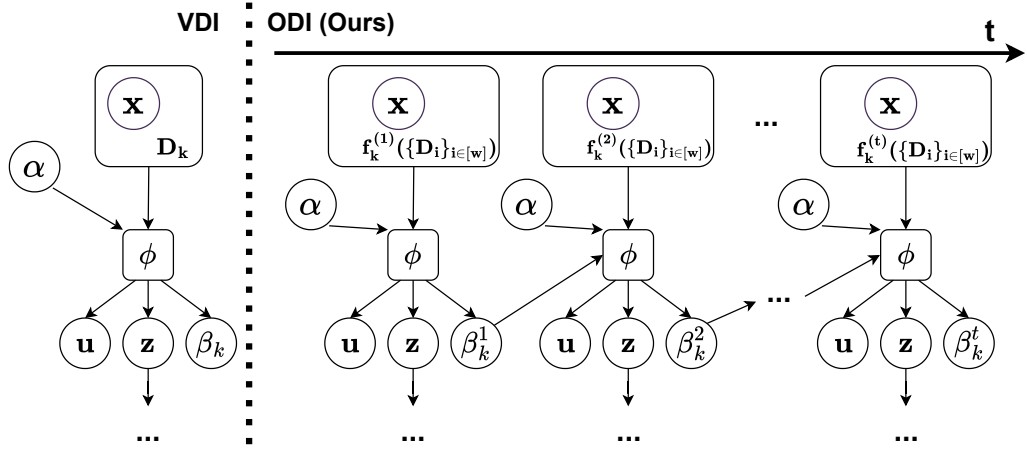

Figure 2: Partial graphical models for VDI (**left**) and our ODI (**right**). Here, $\mathbf{x}$ denotes the input, $\alpha$ is the prior, $\mathbf{u}$ is the local domain index, $\mathbf{z}$ is the data encoding, and $\beta$ is the global domain index. In VDI (**left**), for a domain $k$, the effective sample is $D_k$, while in our ODI (**right**), the effective sample is reweighted as a result of the smoothing function $f_k^t(\{D_i\}_{i \in w})$ in domain $k$ at round $t$. The details for smoothing function are provided in Eqn. (4). In addition, for each round, the global domain index from the previous round $t-1$ also influences the encoder (i.e., $\phi$)'s output. See Fig. 5 in Appendix A for the complete graphical models for VDI and ODI.

global domain indices $\{\boldsymbol{\beta}_k\}_{k=1}^K$. The overall objective is defined as:

$$
\max_{\theta,\phi} \min_{D} \mathcal{L}_{\text{VDI}} = \max_{\theta,\phi} \min_{D} \mathcal{L}_{\theta,\phi} - \lambda_d \mathcal{L}_{D,\phi}
$$

$$
= \max_{\theta,\phi} \min_{D} \mathbb{E}_{p(\mathbf{x},y)}[\mathcal{L}_{\text{ELBO}}(\mathbf{x}, y; \theta, \phi)] - \lambda_d \mathbb{E}_{p(k,\mathbf{x})} \mathbb{E}_{q_\phi(\mathbf{z}|\mathbf{x})}[\log D(k|\mathbf{z})], \quad (1)
$$

where $\lambda_d$ controls the strength of domain invariance through the adversarial discriminator $D$.

Theoretical analysis in (Xu et al., 2023) demonstrates that jointly maximizing the ELBO and minimizing domain classification accuracy via the adversarial discriminator $D$ is equivalent to inferring the optimal global domain index $\boldsymbol{\beta}_k$ for each domain and the optimal local domain index $\mathbf{u}_i$ for each data point $i$. To facilitate discussion of our method in Sec. 3.2, we rewrite the ELBO into two components, one with and one without the domain prior, as Eqn. (2) below:

$$
\mathbb{E}_{p(\mathbf{x},y)}\left[\mathcal{L}_{\text{ELBO}}(\mathbf{x}, y; \theta, \phi)\right] = \mathbb{E}_{p(\mathbf{x},y)}\left[\mathcal{L}_{\text{w/o prior}}(\mathbf{x}, y; \theta, \phi)\right] + \mathbb{E}_{p(\mathbf{x},y)}\left[\mathcal{L}_{\text{prior term}}(\mathbf{x}, y; \theta, \phi)\right] \quad (2)
$$

$$
\mathbb{E}_{p(\mathbf{x},y)}\left[\mathcal{L}_{\text{prior term}}(\mathbf{x}, y; \theta, \phi)\right] = -\mathbb{E}_{q_\phi(\mathbf{u},\boldsymbol{\beta},\mathbf{z}|\mathbf{x})}\left[KL\left[\underbrace{q_\phi(\boldsymbol{\beta}|\mathbf{u})}_{\text{Inferred Domain Index}} \parallel \underbrace{p_\theta(\boldsymbol{\beta}^{(t)}|\boldsymbol{\alpha})}_{\text{Domain Index Prior}}\right]\right] \quad (3)
$$

Fig. 2 shows VDI and our ODI's key components. Please refer to Appendix A for the associated complete probabilistic graphical models.

## 3.2 ONLINE DOMAIN INDEXING (ODI)

**Motivation.** VDI infers domain indices through variational inference and adversarial regularization. However, its effectiveness relies on idealized assumptions – namely, static environments, full domain observability, and balanced domain-wise sample distributions – that rarely hold in practice. In contrast, our setting involves continual domain adaptation under online distribution shift, partial domain observability, and long-tailed domain imbalance (see Fig. 1). These challenges degrade the performance and stability of standard VDI, motivating the design of our *Online Domain Indexing (ODI)*, an extension tailored to address evolving, imbalanced, and partially observed environments.

**Graphical Model of ODI.** To reflect the dynamic and imbalanced nature of our setting, we modify the graphical model of the original VDI. As shown in Fig. 2 (bottom):

1. Our ODI introduces a temporal axis representing training rounds $t$, along which the generative process evolves.

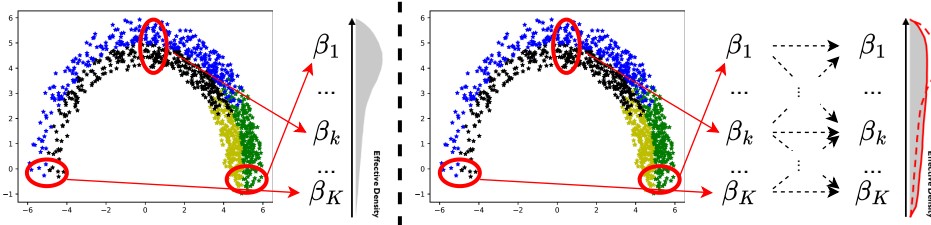

Figure 3: **Left:** In our online DA setting, different domains $k$ contain different numbers of data points. **Right:** Our proposed domain-index-aware reweighting (DIAR) to mitigate such data imbalance.

2. Instead of using a fixed global prior $\alpha$ throughout training, we introduce a time-varying prior where the domain index $\boldsymbol{\beta}_k^{(t)}$ is conditioned on its previous value $\boldsymbol{\beta}_k^{(t-1)}$, thereby encouraging temporal continuity in domain semantics.

3. Meanwhile, the raw domain-level sample counts $\{D_k\}_{k \in [K]}$ used in VDI are replaced by smoothed estimates $f_k^{(t)}(\{D_i\}_{i \in K_w}; w)$ to support more stable reweighting under domain imbalance, where $w$ denotes the size of a window centered around domain $k$, and $K_w = \{i\}_{i=k-\lfloor w/2 \rfloor}^{k+\lfloor w/2 \rfloor} \subseteq [K]$ is the set of neighboring domains in the window.

Below, we detail these **two key innovations of ODI**: (1) temporal priors over domain indices, and (2) domain-index-aware reweighting.

**Temporal Prior on Domain Indices.** VDI employs a global prior $\mathcal{N}(\mu_\alpha, \sigma_\alpha)$ (where we denote $\boldsymbol{\alpha} = (\mu_\alpha, \sigma_\alpha)$) shared across all global domain indices $\beta_k^{(t)}$, enforcing a coherent structure in the latent domain space. This fails to capture the dynamics of evolving domains in online settings. To address this challenge, we introduce a Product-of-Gaussian (PoG) prior (Gales & Airey, 2006) for each domain index $\beta_k^{(t)}$ in round $t$, incorporating both global and temporal structure. Specifically, our PoG prior is the product of two Gaussians:

- A global Gaussian prior $\mathcal{N}(\mu_\alpha, \sigma_\alpha)$, shared across all domains and time steps, to maintain inter-domain consistency, i.e., all domains must align with $\mathcal{N}(\mu_\alpha, \sigma_\alpha)$.
- A time-dependent prior $\mathcal{N}(\mu_{\beta_k^{(t-1)}}, \sigma_{\beta_k^{(t-1)}}^2 \mathbf{I})$, which enforces intra-domain smoothness, i.e., domain $k$'s domain index $\boldsymbol{\beta}_k^{(t)}$ and $\boldsymbol{\beta}_k^{(t-1)}$ in rounds $t$ and $t-1$ should be close.

This PoG formulation regularizes the posterior $q_\phi(\beta_k^{(t)}|\cdot)$ to remain faithful both to global structure and temporal evolution. At initialization ($t = 0$), we set $\beta_k^{(0)}$ to the pre-trained domain indices obtained from the offline dataset $\mathcal{D}_{hist}$. Since the log likelihood for the product of two Gaussians is proportional to two $L_2$ terms, our PoG prior changes Eqn. (3) to:

$$\mathbb{E}_{p^{(t)}(\mathbf{x},y)}[\mathcal{L}_{\text{prior term}}(\mathbf{x}, y; \theta, \phi)]$$

$$= -\mathbb{E}_{q_\phi(\mathbf{u}, \boldsymbol{\beta}^{(t)}, \boldsymbol{\beta}^{(t-1)}, \mathbf{z}|\mathbf{x})}\left[KL\left[q_\phi(\boldsymbol{\beta}^{(t)}|\mathbf{u})||p_\theta(\boldsymbol{\beta}^{(t)}|\boldsymbol{\alpha})\right] + \lambda_t KL\left[q_\phi(\boldsymbol{\beta}^{(t)}|\mathbf{u})||p_\theta(\boldsymbol{\beta}^{(t)}|\boldsymbol{\beta}^{(t-1)})\right]\right],$$

where $\lambda_t$ controls the strength of time-dependent prior.

*Intuition.* Each domain index $\beta_k^{(t)}$ is constrained by a global prior for inter-domain alignment, and a temporal prior for intra-domain smoothness. This dual regularization stabilizes posterior inference across rounds – critical for accurate label prediction.

**Domain-Index-Aware Reweighting.** In our online DA setting, different domains $k$ contain different numbers of data points, with the numbers also changing across different rounds $t$. To mitigate such data imbalance, we propose domain-index-aware reweighting (DIAR). Fig. 3 demonstrates our intuition. Each domain is associated with a latent domain index, and each domain index corresponds to a sample count (i.e., the number of data points observed from that domain $k$ in the current round $t$). A smaller distance between two domain indices indicates a stronger semantic similarity between the corresponding domains. Since the domain indices lie in a continuous latent space of dimension $d_\beta$, we define a Gaussian kernel to estimate the local density of each domain in the current training batch.

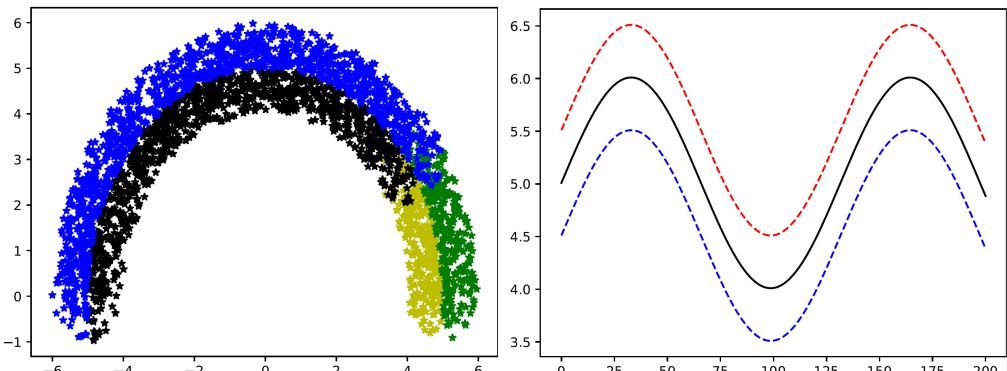

Figure 4: The Growing Circle dataset. **Left:** Visualization of the initial data samples in the 0-th round. Green and blue points represent class 0, while yellow and black points represent class 1. Among them, green and yellow samples are from source domains $\mathcal{S} = \{\mathcal{S}_1, \mathcal{S}_2, \cdots\}$, and blue and black samples are from target domains $\mathcal{T} = \{\mathcal{T}_1, \mathcal{T}_2, \cdots\}$. The circular layout is used to spatially separate source and target domains. **Right:** Visualization of the motion dynamics of the decision boundary and data evolution over time. The black solid line denotes the decision boundary, while the red dashed curve and the blue dashed curve indicate the trajectories of class 0 and class 1 samples, respectively.

In round $t$, for each domain $k_j$, we define its local density kernel over nearby domain indices as:

$$p_{k_j}^{(t)}(k_i) = \frac{1}{\sqrt{2\pi\sigma^2}} \exp\left(-\frac{\text{dist}^2(\boldsymbol{\beta}_{k_i}^{(t)}, \boldsymbol{\beta}_{k_j}^{(t)})}{2\sigma^2}\right),$$

where $k_i \in [K]$ and $\text{dist}(\cdot, \cdot)$ denotes the Euclidean distance in the latent space. Given a fixed window size $w$ for domain $k$, we denote the set of its neighboring domains as $\mathcal{K}_w = \{i\}_{i=k-\lfloor w/2 \rfloor}^{k+\lfloor w/2 \rfloor}$. We compute domain $k$'s smoothed sample count $f_k^{(t)}(\{D_i\}_{i \in [K]}; \mathcal{K}_w)$ as:

$$f_k^{(t)}(\{D_i\}_{i \in \mathcal{K}_w}; \mathcal{K}_w) = \sqrt{\frac{1}{\sum_{i \in K_w} p_k^{(t)}(i)} \sum_{i \in K_w} p_k^{(t)}(i) \cdot D_i^{(t)}}, \tag{4}$$

where $D_i^{(t)}$ denotes the sample count associated with domain $i$ in round $t$. Then we can further reweigh each sample in each domain by the factor of the inverse of such smoothed counts, obtained by a weighted average over neighboring domain counts weighted by its local density. The ELBO in Eqn. (2) then becomes:

$$\mathbb{E}_{p^{(t)}(\mathbf{x}, y, k)}\left[\frac{1}{f_k^{(t)}(\{D_i\}_{i \in \mathcal{K}_w}; \mathcal{K}_w)} \mathcal{L}_{\text{ELBO}}(\mathbf{x}, y; \theta, \phi)\right]. \tag{5}$$

This allows underrepresented domains to borrow statistical strength from adjacent, data-rich domains, smoothing the overall imbalanced domain distribution.

**Maintaining Coverage via Domain-Aware Replay Buffer.** So far, our ODI assumes access to all domains within each training batch in order to compute smoothed domain-level densities. However, in our online setting, only a small subset of domains is accessible in any given round, rendering reweighting infeasible for missing domains.

To address this challenge, we introduce a **domain-aware replay buffer** that maintains a small fixed-size memory for each domain. This buffer stores a representative subset of samples from each domain, allowing the model to approximate full-domain coverage during training. The buffer is updated across training batches by incorporating new data from accessible domains and discarding outdated samples, thereby adapting to the evolving domain distribution over time.

Specifically, we initialize the replay buffer using data from the offline pretraining phase, where we assume balanced and complete domain coverage. For each domain, we randomly select $m$ samples from the pretraining dataset and store them as the initial entries in the buffer. Once online training begins, in each round $t$, we receive a mini-batch of data drawn from a small subset of domains. Let $\mathcal{K}_{access}^{(t)} \subset \mathcal{K}$ denote the set of accessible domains in round $t$. To construct a full-domain training batch, for each inaccessible domain in $\mathcal{K} \setminus \mathcal{K}_{access}^{(t)}$, we augment the current mini-batch with samples

stored in the buffer. Specifically, we retrieve $m$ samples from each corresponding domain buffer and append them to the training batch.

After training on the augmented batch, we update the replay buffer. For each accessible domain $k \in \mathcal{K}_{access}^{(t)}$, we randomly select $m$ samples from the current batch and insert them into the buffer. If the current domain contains fewer than $m$ samples, we overwrite only the first few positions of the buffer queue and retain the remaining samples from previous rounds or batches – effectively applying a FIFO-style update mechanism. This ensures that the buffer remains representative yet up-to-date, and that the smoothing kernel in DIAR consistently has access to all domains during training.

This replay buffer mechanism ensures that all domains remain consistently represented throughout training, enabling reliable reweighting despite partial domain observability.

**Final Objective Function.** Combining the temporal prior on $\boldsymbol{\beta}_k^{(t)}$ and the sample reweighting based on smoothed counts $f_k^{(t)}(\{D_i\}_{i \in K_w}; w)$, we have the following final objective function for round $t$:

$$\max_{\theta,\phi} \min_D \mathcal{L}_{\text{VDI}} = \max_{\theta,\phi} \min_D \mathcal{L}_{\theta,\phi} - \lambda_d \mathcal{L}_{D,\phi}$$

$$= \max_{\theta,\phi} \min_D \left[ \mathbb{E}_{p^{(t)}(\mathbf{x},y,k)} \left[ \frac{1}{f_k^{(t)}(\{D_i\}_{i \in \mathcal{K}_w}; w)} \mathcal{L}_{\text{ELBO}}(\mathbf{x}, y; \theta, \phi) \right] \right.$$

$$\left. - \lambda_d \mathbb{E}_{p^{(t)}(k,\mathbf{x})} \mathbb{E}_{q_\phi(\mathbf{z}|\mathbf{x})} \left[ \frac{1}{f_k^{(t)}(\{D_i\}_{i \in \mathcal{K}_w}; w)} \log D(k|\mathbf{z}) \right] \right], \qquad (6)$$

where $\mathcal{L}_{\text{ELBO}}(\mathbf{x}, y; \theta, \phi)$ is defined as

$$\mathcal{L}_{\text{ELBO}}(\mathbf{x}, y; \theta, \phi) = \mathcal{L}_{\text{w/o prior}}(\mathbf{x}, y; \theta, \phi) - \mathbb{E}_{q_\phi(\mathbf{u},\boldsymbol{\beta}^{(t)},\boldsymbol{\beta}^{(t-1)},\mathbf{z}|\mathbf{x})} \left[ KL \left[ q_\phi(\boldsymbol{\beta}^{(t)}|\mathbf{u}) || p_\theta(\boldsymbol{\beta}^{(t)}|\boldsymbol{\alpha}) \right] \right.$$

$$\left. + \lambda_t KL \left[ q_\phi(\boldsymbol{\beta}^{(t)}|\mathbf{u}) || p_\theta(\boldsymbol{\beta}^{(t)}|\boldsymbol{\beta}^{(t-1)}) \right] \right], \qquad (7)$$

where $\lambda_t$ and $\lambda_d$ are hyper-parameters balancing terms.

## 4 EXPERIMENTS

### 4.1 EXPERIMENTAL SETUP

**Datasets.** We evaluate our proposed ODI framework on three datasets spanning synthetic and real-world scenarios: Growning Circle, CompCars, and TPT-48. These benchmarks are selected to assess robustness under domain shift, partial observability, and domain- or label-wise imbalance. For each dataset, we define the label space, domain structure, and temporal rounds below. To simulate our setting, only a subset of domains is accessible at each round, mimicking partial domain observability. We further apply a Dirichlet distribution to introduce domain-wise sample imbalance by randomly masking samples within each domain per round.

*Growing Circle.* The Growing Circle dataset is a synthetic 2D classification benchmark designed to simulate smooth domain evolution along a circular manifold. It is an evolving version of the Circle dataset used in (Wang et al., 2020; Xu et al., 2023). Fig. 1 shows the dataset in rounds $t = 1$, $t = 50$, and $t = 100$. Each domain corresponds to a specific rotation angle applied to input coordinates (e.g., Domain 1 of Round 1 in Fig. 1(left)), and samples are assigned to one of two classes based on their spatial location. To simulate domain evolution, we control the position of domains over time by adjusting the radius of the circular trajectory. For example, Fig. 1 shows the data distribution gradually shifted from Round $t = 1$ to Round $t = 100$. Specifically, the radius at round $t$ is:

$$r = \sin(\frac{1}{\frac{20}{3}\pi}t) + 5.01$$

As shown in Fig. 4 and Fig. 1, the distribution of samples and decision boundary evolve across rounds. This parameterization allows us to modulate the pace of domain shift over time.

*CompCars.* The CompCars dataset (Yang et al., 2015) consists of over 136,000 real-world car images annotated with attributes such as car type, viewpoint, and year of manufacture (YOM). We define

Table 1: **In-Round Results on Growing Circle.**

| Method | Domain | Avg. Accuracy | | | | | |
|---|---|---|---|---|---|---|---|
| Round $t$ | | Total | [0, 20) | [20, 40) | [40, 60) | [60, 80) | [80, 100) |
| SOURCE-ONLY | N/A | 50.5 | 51.3 | 49.5 | 50.3 | 51.7 | 49.6 |
| DANN | N/A | 49.6 | 49.6 | 48.8 | 49.2 | 50.3 | 50.1 |
| ADDA | N/A | 55.0 | 55.7 | 53.4 | 54.1 | 56.2 | 54.9 |
| CDANN | N/A | 50.8 | 50.2 | 49.9 | 50.1 | 51.3 | 50.5 |
| VDI | N/A | 65.5 | 79.1 | 55.1 | 66.8 | 78.5 | 49.1 |
| VDI-O | 15 | 79.9 | 84.7 | 82.0 | 86.4 | 76.8 | 69.5 |
| ODI (OURS) | 15 | **83.1** | **84.4** | **87.0** | **90.1** | **76.7** | **77.4** |
| VDI-O | 10 | 77.8 | 82.9 | 77.5 | 75.2 | 78.6 | 74.5 |
| ODI (OURS) | 10 | **81.2** | **83.2** | **80.2** | **84.3** | **83.4** | **75.1** |
| VDI-O | 5 | 72.4 | 82.5 | 71.5 | 75.0 | 66.4 | 66.6 |
| ODI (OURS) | 5 | **80.6** | **81.1** | **81.3** | **84.3** | **84.8** | **71.7** |
| UPPER BOUND | 30 | 87.2 | 87.5 | 88.9 | 88.9 | 84.7 | 86.1 |

Table 2: **Next-Round Results on Growing Circle.**

| Method | Domain | Avg. Accuracy | | | | | |
|---|---|---|---|---|---|---|---|
| Round $t$ | | Total | [0, 20) | [20, 40) | [40, 60) | [60, 80) | [80, 100) |
| SOURCE-ONLY | N/A | 50.5 | 51.3 | 49.5 | 50.3 | 51.7 | 49.6 |
| DANN | N/A | 49.6 | 49.6 | 48.8 | 49.2 | 50.3 | 50.1 |
| ADDA | N/A | 55.0 | 55.7 | 53.4 | 54.1 | 56.2 | 54.9 |
| CDANN | N/A | 50.8 | 50.2 | 49.9 | 50.1 | 51.3 | 50.5 |
| VDI | N/A | 65.5 | 79.1 | 55.1 | 66.8 | 78.5 | 49.1 |
| VDI-O | 15 | 80.7 | 85.1 | 81.9 | 87.0 | 79.1 | 70.4 |
| ODI (OURS) | 15 | **83.9** | **84.2** | **86.8** | **90.9** | **79.1** | **78.3** |
| VDI-O | 10 | 78.6 | 82.6 | 78.0 | 76.4 | 80.8 | 75.4 |
| ODI (OURS) | 10 | **82.3** | **84.5** | **80.5** | **85.3** | **85.3** | **76.1** |
| VDI-O | 5 | 72.9 | 81.8 | 71.8 | 75.1 | 68.5 | 67.3 |
| ODI (OURS) | 5 | **81.5** | **82.5** | **81.7** | **84.8** | **85.9** | **72.4** |
| UPPER BOUND | 30 | 88.0 | 87.8 | 88.9 | 89.4 | 86.8 | 87 |

domains by viewpoints (front, rear, side, front-side, rear-side), temporal rounds by years (2009–2014), and labels by car types (MPV, SUV, sedan, hatchback). This setup reflects a realistic domain shift across both visual perspectives and time.

*TPT-48.* The TPT-48 dataset contains monthly average temperatures for the 48 contiguous U.S. states from 1948 to 2019. The raw data come from NOAA's nClimDiv and nClimGrid datasets (Vose et al., 2014), and we use the processed version released by the Washington Post (WP, 2020). We treat each state as a domain, divide the timeline into 12-year rounds, and define the prediction task as forecasting average temperatures for six months from the past six month.

**Baselines.** We compare our proposed *ODI* against a broad set of domain adaptation (DA) baselines, covering both standard and state-of-the-art methods across classification and regression tasks. These include

- **DANN** (Ganin et al., 2016), a seminal adversarial domain adaptation method that promotes domain-invariant representations via a gradient reversal layer.

- **ADDA** (Tzeng et al., 2017), which separately pretrains a source encoder and then adversarially aligns a target encoder to the source feature space.

- **CDANN** (Zhao et al., 2017), an extension of DANN that conditions domain discrimination on class predictions to mitigate class-conditional shift.

- **VDI** (Xu et al., 2023), which formally introduces to infer domain index to further improve domain adaptation performance. We also employ **VDI** (Xu et al., 2023) as the backbone model of our ODI.

**Evaluation Protocols and Metrics.** We report **classification accuracy** for the Growing Circle and CompCars datasets, and **mean squared error (MSE)** for the TPT-48 dataset, reflecting the nature of the prediction task on each dataset. Unlike conventional domain adaptation settings where the model is trained offline and evaluated once on a held-out target domain, our method operates in an **online adaptation** setting. At each round $t$, we conduct two types of evaluation: (1) *in-round evaluation*, where the model trained on round-$t$ data is tested on the target domains within the same round; and (2) *next-round generalization*, where the model trained on round-$t$ is evaluated on the target domains of round $t + 1$. This protocol reflects both adaptation capability under partially observed, imbalanced data and the ability to generalize across temporal shifts in domain distribution.

## 4.2 RESULTS

We evaluate all methods using **accuracy** on the Growing Circle and CompCars datasets, and **mean squared error (MSE)** on the TPT-48 dataset. Results are presented in Table 1~6,

Across all tables, we observe that traditional DA baselines performs poorly as the data distribution evolves since these offline methods fail to capture the temporal and structural shifts across domains and rounds. To further examine whether VDI can be directly applied in online settings, we adapt it into an online version (*VDI-O*) by updating the model offline over rounds. However, without explicitly addressing partial domain observability and domain-wise sample imbalance, *VDI-O* struggles to generalize.

Table 3: **In-Round Results on CompCars.**

| Method | Domain | Avg. Accuracy | | | | | |
|---|---|---|---|---|---|---|---|
| Round $t$ | | Total | 1 | 2 | 3 | 4 | 5 |
| SOURCE-ONLY | N/A | 31.5 | 34.9 | 33.1 | 28.7 | 29.3 | 31.4 |
| DANN | N/A | 34.9 | 37.4 | 36.7 | 32.3 | 33.0 | 35.1 |
| ADDA | N/A | 41.1 | 43.9 | 41.4 | 40.4 | 39.1 | 39.6 |
| CDANN | N/A | 37.0 | 38.6 | 38.0 | 35.4 | 36.1 | 36.9 |
| VDI | N/A | 41.2 | 45.6 | 43.3 | 38.3 | 38.4 | 40.3 |
| VDI-O | 2 | 39.2 | 46 | 39.5 | 34.8 | 41.6 | 34.1 |
| ODI (OURS) | 2 | **44.3** | **48.4** | **45.2** | **44.0** | **41.9** | **41.9** |
| UPPER BOUND | 5 | 46.1 | 47.8 | 44.2 | 46.3 | 46.9 | 45.2 |

Table 4: **Next-Round Results on CompCars.**

| Method | Domain | Avg. Accuracy | | | | | |
|---|---|---|---|---|---|---|---|
| Round $t$ | | Total | 1 | 2 | 3 | 4 | 5 |
| SOURCE-ONLY | N/A | 31.5 | 34.9 | 33.1 | 28.7 | 29.3 | 31.4 |
| DANN | N/A | 34.9 | 37.4 | 36.7 | 32.3 | 33.0 | 35.1 |
| ADDA | N/A | 41.1 | 43.9 | 41.4 | 40.4 | 39.1 | 39.6 |
| CDANN | N/A | 37.0 | 38.6 | 38.0 | 35.4 | 36.1 | 36.9 |
| VDI | N/A | 41.2 | 45.6 | 43.3 | 38.3 | 38.4 | 40.3 |
| VDI-O | 2 | 38.6 | 45.7 | 40.0 | 35.6 | 33.0 | 38.6 |
| ODI (OURS) | 2 | **42.1** | **45.7** | **43.9** | **41.2** | **40.0** | **40.6** |
| UPPER BOUND | 5 | 42.9 | 45.7 | 44.0 | 42.6 | 40.6 | 41.9 |

Table 5: **In-Round Results on TPT-48.**

| Method | Domain | Avg. MSE | | | | | |
|---|---|---|---|---|---|---|---|
| Round $t$ | | Total | 1 | 2 | 3 | 4 | 5 |
| SOURCE-ONLY | N/A | 8.59 | 8.62 | 8.23 | 8.60 | 8.87 | 8.39 |
| DANN | N/A | 8.61 | 8.64 | 8.47 | 8.68 | 8.76 | 8.44 |
| ADDA | N/A | 8.68 | 8.72 | 8.53 | 8.70 | 8.95 | 8.45 |
| CDANN | N/A | 8.69 | 8.66 | 8.54 | 8.81 | 8.90 | 8.50 |
| VDI | N/A | 7.09 | 6.89 | 7.05 | 7.21 | 7.35 | 6.95 |
| VDI-O | 24 | 6.63 | 6.99 | 6.62 | 6.27 | 6.49 | 6.83 |
| ODI (OURS) | 24 | **6.07** | **6.49** | **5.92** | **5.89** | **6.11** | **5.99** |
| UPPER BOUND | 48 | 5.92 | 6.43 | 5.76 | 5.67 | 5.93 | 5.79 |

Table 6: **Next-Round Results on TPT-48.**

| Method | Domain | Avg. MSE | | | | | |
|---|---|---|---|---|---|---|---|
| Round $t$ | | Total | 1 | 2 | 3 | 4 | 5 |
| SOURCE-ONLY | N/A | 8.59 | 8.62 | 8.23 | 8.60 | 8.87 | 8.39 |
| DANN | N/A | 8.61 | 8.64 | 8.47 | 8.68 | 8.76 | 8.44 |
| ADDA | N/A | 8.68 | 8.72 | 8.53 | 8.70 | 8.95 | 8.45 |
| CDANN | N/A | 8.69 | 8.66 | 8.54 | 8.81 | 8.90 | 8.50 |
| VDI | N/A | 7.09 | 6.89 | 7.05 | 7.21 | 7.35 | 6.95 |
| VDI-O | 24 | 6.86 | 7.19 | 6.89 | 7.04 | 6.53 | 6.66 |
| ODI (OURS) | 24 | **6.42** | **6.89** | **6.72** | **6.49** | **5.92** | **6.09** |
| UPPER BOUND | 48 | 6.18 | 6.89 | 6.60 | 6.20 | 5.49 | 5.71 |

In contrast, our proposed method **ODI** consistently outperforms all baselines. By incorporating temporal priors over domain indices and kernel-based reweighting to correct for domain imbalance, ODI enables smooth inter-round adaptation and achieves stronger predictive performance under realistic constraints.

We also report an *Upper Bound* (Oracle) baseline, in which *VDI-O* has access to all domain data with perfectly balanced distributions at each round. Although impractical in real-world deployments, this serves as a performance ceiling (upper bound). Results show that our ODI can successfully achieve performance very close to this upper bound, verifying its effectiveness.

## 5 LIMITATION

While our method provides a strong solution under an extreme and realistic continual adaptation setting, several limitations remain. First, the effectiveness of our approach relies on the assumption of smooth domain shifts across rounds; when domain shifts are abrupt (e.g., sudden design changes in CompCars), the outdated buffer samples may no longer provide useful guidance, hurting performance. Second, our reweighting strategy uses a simple global Gaussian kernel, which may be suboptimal under highly irregular domain layouts.

## 6 CONCLUSION

In this paper, we propose **ODI**, a novel framework for online domain adaptation under evolving, partially observable, and imbalanced domain-wise distributions. Our method extends the Variational Domain Index (VDI) framework by incorporating *temporal priors* to ensure inter-round domain consistency, *kernel-based smoothing* to handle domain-wise sample imbalance, and a *domain-aware replay buffer* to mitigate partial domain visibility. Together, these innovations enable stable and effective adaptation in settings where both domains and distributions evolve over time – a scenario commonly overlooked by existing methods. Extensive experiments on both synthetic and real-world datasets demonstrate that Imbalanced VDI outperforms state-of-the-art baselines under challenging continual adaptation settings. Future work includes developing adaptive smoothing kernels, exploring more principled buffer update strategies, and extending the framework to high-resolution vision or multi-modal applications.

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

## A  COMPLETE GRAPHICAL MODELS

The complete graphical models of VDI and our ODI are shown in Fig. 5.

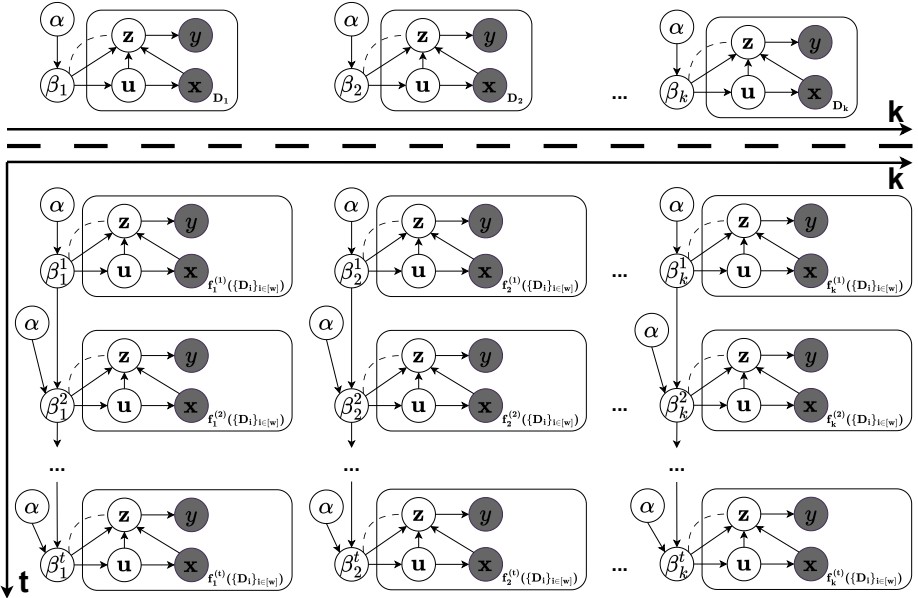

Figure 5: Complete graphical models of VDI (**top**) and our ODI (**bottom**).

## B  LLM USAGE

We employed large language models (LLMs) exclusively to enhance the presentation of this paper, focusing on grammar, clarity, and overall readability. The LLMs were not involved in generating ideas, designing experiments, conducting analyses, or producing any scientific content. All research contributions, technical claims, and conclusions are entirely the work of the authors.

## C  DISCUSSION

### C.1  DETAILS FOR DATASETS

We follows the literature (e.g., VDI (Xu et al., 2023)) to use CompCars, TPT-48, and an ODI version of the Circle dataset. These datasets are not small-scaled. Specifically,

- **Circle -> Growing Circle:** In VDI, it only consists of the data before the online stage, however, in our ODI settings, we include $T + 1$ times more than sample used in VDI due to the domain evolution through round.
- **CompCars:** We directly follow the dataset used in the VDI paper, where there are 18,735 images in total across $5 \times 6 = 30$ domains..
- **TPT-48:** In VDI, it only consists of the data before the online stage, which consists of 6,384 data in total across $48$ domains. However, in our ODI settings, we include $T = 6$ times more than sample used in VDI due to the domain evolution through round, which results in 38,304 data across $6 \times 48 = 288$ domains.

### C.2  DETAILS FOR EXPERIMENTAL IMPLEMENTATIONS

**Distributional Shift (Online Evolution Across Time).** We simulate an online environment with $T$ rounds and $K$ domains. As time progresses, each domain's distribution evolves, instead of remaining fixed as in traditional DA settings (which typically assume a single source and a single target). This

produces **continuous temporal drift** within each domain, naturally inducing an online distributional shift. Note that in our setting such shift happens for all $K$ domains, as shown in Fig. 1 of the paper.

**Partial Observability (Only a Subset of Domains Available at Each Time).** Although $K$ domains exist globally, in realistic scenarios not all domains provide data at every round. Therefore, at each round $t$, we randomly sample \*\*only a small subset\*\* of domains to be observed (e.g., 2–3 out of 10), while the remaining domains contribute no data. This mimics real-world situations where data from some domains arrive with delay or are temporarily missing, resulting in partial observability.

**Data Sparsity and Imbalanced Access.** Even among the domains observed at round $t$, the number of available samples is highly imbalanced. While an ideal scenario might yield a balanced allocation (e.g., 100 samples per domain), in practice some domains may contribute 80 samples, others only 10 or even fewer. This creates severe data sparsity and domain-level imbalance, which ODI is specifically designed to address.

### C.3 MOTIVATING EXAMPLE

A typical example would be social networking platforms like Instagram, where users from different countries post photos and videos, and personalized ad recommendations are made based on their activities. Due to time zone differences, only a fraction of countries have high levels of user activity at any given time. For example, when users in the US are active, users in Asia are asleep, and vice versa. Here, each country corresponds to one domain, and a global social platform like Instagram clearly needs to handle multi-domain data. The distribution of such multi-domain data evolves over time due to time zone differences between countries, making our setting realistic. Another example is, climate or environmental sensor networks deployed across widely separated regions form multiple domains whose data do not arrive synchronously. Nearby regions may provide timely data, while distant regions (e.g., collecting African climate signals from a U.S.-based server) may exhibit substantial delays. Thus, at each time step, only a partially observed and highly imbalanced subset of domains is available (e.g., 2–3 out of 10), while all domains evolve over time due to environmental changes. This multi-domain, time-varying, partially observable structure aligns naturally with our ODI formulation and goes beyond the assumptions of traditional domain adaptation.

### C.4 CLARIFICATIONS FOR OUR PROBLEM SETTING

#### C.4.1 TRADITIONAL SETTING

We would like to clarify that our ODI setting is very different from typical settings; therefore a lot of methods are not directly applicable. Specifically:

- In the traditional DA setting, one usually assumes $K$ continuous domains. This is a **one-dimensional** domain structure.
- In our ODI setting, we assume $K$ continuous domains **for each time step** $t \in \{1, \ldots, T\}$. This is a **two-dimensional** domain structure. For example, with 10 domains across 20 time steps, the data consists of total 200 domains.

Each domain evolves over time, and different domains also evolve differently. Thus, ODI involves **jointly time-varying and domain-varying distributions**.

Our ODI method is specifically designed for this much more complex, multi-domain, time-evolving scenario, and therefore is **not applicable** to the traditional DA setting, and methods like CDA is also not directly applicable to our ODI setting. Note that our **main baseline VDI** is **not applicable** to the traditional DA setting **either**.

#### C.4.2 CONSTRAINTS IN OUR SETTINGS

We would like to clarify that our problem setting incorporates both dynamic and resource-constrained conditions:

**Dynamic setting.** Our ODI setting is inherently dynamic in three ways:

- Dynamic distributions: At each time step $t$, all domains $d_1, d_2, ..., d_{10}$ evolve over time (e.g., across 20 time steps). Thus each domain has a temporally shifting distribution.

- Dynamic partial observability: Although 10 domains exist, only a random subset (e.g., 3–5 domains) is observed at each time step, making the domain stream partially observable.

- Dynamic imbalance: Even among the observed domains, the available samples are extremely imbalanced, further increasing the difficulty of adaptation.

These elements together create a multi-domain, time-evolving, partially observed, imbalanced environment that is much more dynamic than classical DA settings.

**Resource-constrained setting.** If resources were unlimited, one could simply aggregate all historical data and retrain the model from scratch at every time step. This would largely mitigate imbalance and instability. However, in realistic scenarios, as the number of time steps grows, continuously scaling the training set is computationally prohibitive, making full retraining infeasible. This is precisely why we introduce the domain-aware replay buffer: we selectively retain a subset of past samples and perform online adaptation on top of them, enabling effective domain adaptation under strict resource limitations.

## D  REWEIGHTING NOVELTY

This is a good question. We would like to clarify that our continuous domain reweighting techniques are different from existing discrete domain reweighting techniques. Specifically:

- **Existing Discrete Domain Reweighting versus Our Continuous Domain Reweighting.** Conventional approaches treat domains as discrete, isolated categories and assign weights independently to each domain (e.g., weighting domain solely based on the empirical statistics within domain $k$). Such methods implicitly assume that domains have no intrinsic ordering or geometric structure. However, in our setting, domains lie on a *continuous* index $\beta$, and neighboring domains share latent generative factors and exhibit smooth transitions in distribution. Treating these domains as discrete entities fails to exploit this structure and leads to unstable or overly sharp weighting profiles, particularly when some domains have limited sample support.

- **Existing Discrete Domain Reweighting versus Our Continuous Domain Reweighting.** Conventional approaches treat domains as discrete, isolated categories and assign weights independently to each domain (e.g., weighting domain solely based on the empirical statistics within domain $k$). Such methods implicitly assume that domains have no intrinsic ordering or geometric structure. However, in our setting, domains lie on a *continuous* index $\beta$, and neighboring domains share latent generative factors and exhibit smooth transitions in distribution. Treating these domains as discrete entities fails to exploit this structure and leads to unstable or overly sharp weighting profiles, particularly when some domains have limited sample support. Our method instead leverages the continuity of the domain index to perform **continuous domain reweighting**, where the weight assigned to domain $\beta$ depends not only on its own empirical statistics but also on those of *its neighboring domains*. Mathematically, we construct a smoothed effective frequency function $f_{\text{eff}}(\beta)$ via kernel-based convolution over the raw domain frequencies. Reweighting is then applied using this continuous estimate, yielding a weight function $w(\beta) \propto 1/f_{\text{eff}}(\beta)$ that varies smoothly and respects the underlying domain geometry. This produces a more stable, data-efficient, and theoretically coherent weighting scheme that cannot be replicated by discrete reweighting heuristics.

- **Why Discrete Reweighting Cannot Capture the Continuous Structure.** Discrete reweighting approaches implicitly treat all domain boundaries as discontinuities; even infinitesimally adjacent domains receive completely independent weights. As a result, small fluctuations in domain-level sample counts can lead to large jumps in the assigned weights. In contrast, our method enforces smoothness consistent with the assumption that the domain index evolves gradually (e.g., via time zones, temporal progression, environmental drift, or geographic continuity). This alignment with the problem structure is crucial: it reduces variance, mitigates overfitting to sampling noise, and enables principled extrapolation to underrepresented or sparsely sampled regions of the domain index.

Table 7: **Ablation Studies of In-round accuracy on CompCars.**

| Method | Domain | Avg. Accuracy | | | | | |
|---|---|---|---|---|---|---|---|
| Round $t$ | | Total | 1 | 2 | 3 | 4 | 5 |
| VDI-O | 2 | 39.2 | 46 | 39.5 | 34.8 | 41.6 | 34.1 |
| ODI w/o P | 2 | 43.3 | 46.7 | 44.1 | 42.8 | 41.4 | 41.3 |
| ODI w/o B | 2 | 40.0 | 45.6 | 39.4 | 37.4 | 41.8 | 35.4 |
| ODI w/o R | 2 | 40.4 | 45.0 | 44.3 | 35.7 | **42.6** | 34.4 |
| ODI (OURS) | 2 | **44.3** | **48.4** | **45.2** | **44.0** | 41.9 | **41.9** |
| UPPER BOUND | 5 | 46.1 | 47.8 | 44.2 | 46.3 | 46.9 | 45.2 |

Table 8: **Ablation Studies of Next-round accuracy on CompCars.**

| Method | Domain | Avg. Accuracy | | | | | |
|---|---|---|---|---|---|---|---|
| Round $t$ | | Total | 1 | 2 | 3 | 4 | 5 |
| VDI-O | 2 | 38.6 | 45.7 | 40.0 | 35.6 | 33.0 | 38.6 |
| ODI w/o P | 2 | 40.8 | 44.1 | 41.6 | 40.7 | 38.2 | 39.5 |
| ODI w/o B | 2 | 40.4 | 45.3 | 40.8 | 40.2 | 36.9 | 38.8 |
| ODI w/o R | 2 | 40.2 | 45.7 | 40.4 | 41.1 | 33.7 | 39.9 |
| ODI (OURS) | 2 | **42.1** | **45.7** | **43.9** | **41.2** | **40.0** | **40.6** |
| UPPER BOUND | 5 | 42.9 | 45.7 | 44.0 | 42.6 | 40.6 | 41.9 |

Table 9: **In-Round Results on CompCars.**

| Method | Domain | Avg. Accuracy | | | | | |
|---|---|---|---|---|---|---|---|
| Round $t$ | | Total | 1 | 2 | 3 | 4 | 5 |
| VDI-O | 2 | 39.2 | 46 | 39.5 | 34.8 | 41.6 | 34.1 |
| CDA | 2 | 40.6 | 44.9 | 43.0 | 37.4 | 38.5 | 39.1 |
| ODI (OURS) | 2 | **44.3** | **48.4** | **45.2** | **44.0** | 41.9 | **41.9** |
| UPPER BOUND | 5 | 46.1 | 47.8 | 44.2 | 46.3 | 46.9 | 45.2 |

Table 10: **Next-Round Results on CompCars.**

| Method | Domain | Avg. Accuracy | | | | | |
|---|---|---|---|---|---|---|---|
| Round $t$ | | Total | 1 | 2 | 3 | 4 | 5 |
| VDI-O | 2 | 38.6 | 45.7 | 40.0 | 35.6 | 33.0 | 38.6 |
| CDA | 2 | 40.3 | 44.6 | 42.9 | 37.1 | 38.0 | 38.7 |
| ODI (OURS) | 2 | **42.1** | **45.7** | **43.9** | **41.2** | **40.0** | **40.6** |
| UPPER BOUND | 5 | 42.9 | 45.7 | 44.0 | 42.6 | 40.6 | 41.9 |

- **Empirical and Practical Implications.** The benefit of continuous reweighting is most pronounced in real-world multi-domain datasets where the domain index evolves smoothly (e.g., temporal shifts or spatial differences). In such settings, discrete reweighting often produces unstable training dynamics or suboptimal performance due to noisy or imbalanced domain counts. Our continuous method offers a principled alternative that yields more robust gradient estimates, better utilization of cross-domain information, and improved generalization performance, as confirmed by our experimental results.

# E  ADDITIONAL RESULTS

## E.1  ABLATION STUDIES

We reported ablation studies for (1) temporal priors **(P)**, (2) domain-index-aware reweighting **(R)**, and (3) replay buffer **(B)** on the CompCars dataset with 5 time steps (i.e., 5 temporal rounds), Table 7 and Table 8 summarizes our new results.

As shown in Table 7 and Table 8, the full ODI model consistently outperforms all its ablated variants across both in-round and next-round evaluations. These results highlight the complementary roles of all three components:

- **Temporal prior (P)** significantly improves stability and carries useful information forward across rounds. Removing it (ODI w/o P) leads to noticeable drops in both current- and next-round accuracy.

- **Domain-index-aware reweighting (R)** is essential for handling domain-wise imbalance. Without it (ODI w/o R), the model becomes overly biased toward the majority domains and fails to generalize across rounds.

- **Replay buffer (B)** is particularly important under partial observability. Removing it (ODI w/o B) harms the model's ability to retain information about previously unseen or lightly sampled domains.

Importantly, ODI outperforms VDI-O by a clear margin, demonstrating that simply applying an offline DA method (VDI-O) to the online setting is insufficient. All three online components, i.e., temporal priors, domain-index-aware reweighting, and replay buffering, are necessary to handle distributional shift, partial observability, and domain imbalance in online environments.

### E.2 ADDITIONAL BASELINES

We adapt Continuous Domain Adaptation (CDA) (Xu et al., 2022a) to our setting and ran additional experiments to evaluate it. Specifically, we fixed 2 accessible domains per round on the CompCars dataset with 5 time steps (i.e., 5 temporal rounds), Table 9 and Table 10 summarizes our new results.

