# OpenReview forum: "Online Domain Indexing"
_ICLR.cc/2026/Conference — Submitted to ICLR 2026_

### Official Review · Reviewer_21zP · 2025-10-31

**Soundness:** 2
**Presentation:** 3
**Contribution:** 2
**Rating:** 4
**Confidence:** 5

**Summary:**

The method tackles the problem of existing DA methods, which assumes that all data can be accessed with balanced distribution. However, in most applications, data are accessed online and in order, where only partial domains’ data are seen and accessible. They proposed the online-domain indexing framework, with the online-imbalanced DA problem, where only a few domains can be accessed with imbalanced data number. The proposed contributions incorporate the temporal prior on domain indices, domain-index-aware-reweighting, and the domain-aware replay buffer.

**Strengths:**

-	The new problem seems sound and from the real-world application scenarios.
-	The optimization and the lower bound seems correct.
-	A toy-example dataset is generated to prove the method and theory.

**Weaknesses:**

-	This method seems to integrate the partially observable (continual DA) with the VDI. The complexity of the setup seems unrealistic.
-	The method wise, it seems that the authors missed another line of works, focusing on the ``index-less’’ continuous domain adaptation [r1]. This paper seems to consider the continually coming data but without knowing the domain index. It seems a good solution to the proposed setup and also the simplified setup.
-	The evaluation is only on the small-scale dataset; The comparison methods are very limited.
-	Reweighting techniques have been utilized and applied in DA for a long time. The overall novelty seems limited by integrating several techniques together.

[r1] Delving into the Continuous Domain Adaptation, MM, 2022

**Questions:**

See weakness.

---

> ### Author Response · Authors · 2025-11-21
> **[1/3] Rebuttal**
>
> Thank you for your constructive comments. We are glad that you find our problem `"sound and from the real-world application scenarios"`, our optimization and lower bound `"seems correct"`. Below, we address your questions one by one in detail. We have also **included all discussions below in our revision** (with the changed part marked in blue).
>
>
> **Q1: "This method seems to integrate the partially observable (continual DA) with the VDI. The complexity of the setup seems unrealistic."**
>
> Thank you for your concern about the setting. We would like to clarify that our setting is motivated by real-world production scenarios. We are currently working with a major global AI company on deploying this framework, and such multi-domain, time-evolving structures naturally arise in applications such as online advertising systems, where user groups (domains) and their distributions shift concurrently over time.
>
> A typical example would be social networking platforms like Instagram, where users from different countries post photos and videos, and personalized ad recommendations are made based on their activities. Due to time zone differences, only a fraction of countries have high levels of user activity at any given time. For example, when users in the US are active, users in Asia are asleep, and vice versa. Here, each country corresponds to one domain, and a global social platform like Instagram clearly needs to handle multi-domain data. The distribution of such multi-domain data evolves over time due to time zone differences between countries, making our setting realistic.
>
> We will add this clarification to the revised version of the paper.
>
> **Q2: "The method wise, it seems that the authors missed another line of works, focusing on the ``index-less’’ continuous domain adaptation [r1]. This paper seems to consider the continually coming data but without knowing the domain index. It seems a good solution to the proposed setup and also the simplified setup."**
>
> Thank you for pointing us to this interesting paper, which we will cite and discuss in the revision.
>
> We would also like to clarify that our ODI setting is fundamentally different from the setting used in that work, and therefore the two setups are not directly comparable.
> + In the Continuous Domain Adaptation (CDA) setting [r1], they assume $K$ continuous domains. This is a **one-dimensional** domain structure.
> + In our ODI setting, we assume $K$ continous domains **for each time step $t\in \{1,\dots, T\}$**. This is a **two-dimensional** domain structure. For example, with 10 domains across 20 time steps, the data consists of total 200 domains.
>
> | t | $D_1$ | $D_2$ | $D_3$ | ... | $D_{10}$ |
> |---|-----|-----|-----|-----|-----|
> |1| $D_{1}^{1}$ | $D_{2}^{1}$ | $D_{3}^{1}$ | ... | $D_{10}^{1}$ |
> |2| $D_{1}^{2}$ | $D_{2}^{2}$ | $D_{3}^{2}$ | ... | $D_{10}^{2}$ |
> |3| $D_{1}^{3}$ | $D_{2}^{3}$ | $D_{3}^{3}$ | ... | $D_{10}^{3}$ |
> |...| ... | ... | ... | ... |... |
> |20| $D_{1}^{20}$ | $D_{2}^{20}$ | $D_{3}^{20}$ | ... | $D_{10}^{20}$ |
>
> Each domain evolves over time, and different domains also evolve differently. Thus, ODI involves **jointly time-varying and domain-varying distributions**.
>
> Due to such differences, CDA is not directly applicable to our ODI setting.
>
> Nevertheless, following your suggestion, we made every effort to adapt Continuous Domain Adaptation (CDA) [r1] to our setting and ran additional experiments to evaluate it. Specifically, we fixed $2$ accessible domains per round on the CompCar dataset with 5 time steps (i.e., 5 temporal rounds), and Table A & B below summarizes our new results.
>
> Table A. In-Round Results on CompCars
>
> | Model | 1 | 2 | 3 | 4 | 5 | Average |
> |------------------|---------:|-------:|-------:|------:|--------:|---------:|
> | **VDI-O** | 46.0 | 39.5 | 34.8 |  41.6 | 34.1 | 39.2 |
> | **CDA** | 44.9 | 43.0 | 37.4 | 38.5 | 39.1 | 40.6 |
> | **ODI** | **48.4** | **45.2** | **44.0** | **41.9** | **41.9** | **44.3** |
>
> Table B. Next-Round Results on CompCars
>
> | Model | 1 | 2 | 3 | 4 | 5 | Average |
> |------------------|---------:|-------:|-------:|------:|--------:|---------:|
> | **VDI-O** | **45.7** | 40.0 | 35.6 |  33.0 | 38.6 | 38.6 |
> | **CDA** | 44.6 | 42.9 | 37.1 | 38.0 | 38.7 | 40.3 |
> | **ODI** | **45.7** | **43.9** | **41.2** | **40.0** | **40.6** | **42.1** |
>
> We have cited this paper in the Related Work section and included the new results in the Appendix of the revised version as suggested.

---

> ### Author Response · Authors · 2025-11-21
> **[2/3] Rebuttal**
>
> **Q3.1: "The evaluation is only on the small-scale dataset; The comparison methods are very limited."**
>
> Thank you mentioning this. Actually our paper follows the literature (e.g., VDI [1]) to use CompCars, TPT-48, and an ODI version of the Circle dataset. These datasets are not small-scaled. Specifically,
>
> - **Circle -> Growing Circle:** In VDI, it only consists of the data before the online stage, however, in our ODI settings, we include $T+1$ times more than sample used in VDI due to the domain evolution through round.
> - **CompCars:** We directly follow the dataset used in the VDI paper, where there are 18,735 images in total across $5 \times 6 = 30$ domains.
> - **TPT-48:** In VDI, it only consists of the data before the online stage, which consists of 6,384 data in total across $48$ domains. However, in our ODI settings, we include $T=6$ times more than sample used in VDI due to the domain evolution through round, which results in 38,304 data across $6 \times 48 = 288$ domains.
>
>
> **Q3.2: "The comparison methods are very limited."**
>
> We would like to clarify that our ODI setting is very different from typical settings; therefore a lot of methods are not directly applicable. Specifically:
>
> + In the Continuous Domain Adaptation (CDA) setting [r1], they assume $K$ continuous domains. This is a **one-dimensional** domain structure.
> + In our ODI setting, we assume $K$ continous domains **for each time step $t\in \{1,\dots, T\}$**. This is a **two-dimensional** domain structure. For example, with 10 domains across 20 time steps, the data consists of total 200 domains.
>
> | t | $D_1$ | $D_2$ | $D_3$ | ... | $D_{10}$ |
> |---|-----|-----|-----|-----|-----|
> |1| $D_{1}^{1}$ | $D_{2}^{1}$ | $D_{3}^{1}$ | ... | $D_{10}^{1}$ |
> |2| $D_{1}^{2}$ | $D_{2}^{2}$ | $D_{3}^{2}$ | ... | $D_{10}^{2}$ |
> |3| $D_{1}^{3}$ | $D_{2}^{3}$ | $D_{3}^{3}$ | ... | $D_{10}^{3}$ |
> |...| ... | ... | ... | ... |... |
> |20| $D_{1}^{20}$ | $D_{2}^{20}$ | $D_{3}^{20}$ | ... | $D_{10}^{20}$ |
>
> Each domain evolves over time, and different domains also evolve differently. Thus, ODI involves **jointly time-varying and domain-varying distributions**.
>
> Due to such differences, methods like CDA is not directly applicable to our ODI setting. We are more than happy to include additional baselines you suggest; please feel free to let us know if you have any particular method in mind.

---

> ### Author Response · Authors · 2025-11-21
> **[3/3] Rebuttal**
>
> **Q4: "Reweighting techniques have been utilized and applied in DA for a long time. The overall novelty seems limited by integrating several techniques together."**
>
> This is a good question. We would like to clarify that our continuous domain reweighting techniques are different from existing discrete domain reweighting techniques. Specifically:
>
> **Existing Discrete Domain Reweighting versus Our Continuous Domain Reweighting.** Conventional approaches treat domains as discrete, isolated categories and assign weights independently to each domain (e.g., weighting domain solely based on the empirical statistics within domain $k$). Such methods implicitly assume that domains have no intrinsic ordering or geometric structure. However, in our setting, domains lie on a *continuous* index $\beta$, and neighboring domains share latent generative factors and exhibit smooth transitions in distribution. Treating these domains as discrete entities fails to exploit this structure and leads to unstable or overly sharp weighting profiles, particularly when some domains have limited sample support.
>
> Our method instead leverages the continuity of the domain index to perform **continuous domain reweighting**, where the weight assigned to domain $\beta$ depends not only on its own empirical statistics but also on those of *its neighboring domains*. Mathematically, we construct a smoothed effective frequency function $f_{\mathrm{eff}}(\beta)$ via kernel-based convolution over the raw domain frequencies. Reweighting is then applied using this continuous estimate, yielding a weight function $w(\beta) \propto 1 / f_{\mathrm{eff}}(\beta)$ that varies smoothly and respects the underlying domain geometry. This produces a more stable, data-efficient, and theoretically coherent weighting scheme that cannot be replicated by discrete reweighting heuristics.
>
> **Why Discrete Reweighting Cannot Capture the Continuous Structure.** Discrete reweighting approaches implicitly treat all domain boundaries as discontinuities; even infinitesimally adjacent domains receive completely independent weights. As a result, small fluctuations in domain-level sample counts can lead to large jumps in the assigned weights. In contrast, our method enforces smoothness consistent with the assumption that the domain index evolves gradually (e.g., via time zones, temporal progression, environmental drift, or geographic continuity). This alignment with the problem structure is crucial: it reduces variance, mitigates overfitting to sampling noise, and enables principled extrapolation to underrepresented or sparsely sampled regions of the domain index.
>
> **Empirical and Practical Implications.** The benefit of continuous reweighting is most pronounced in real-world multi-domain datasets where the domain index evolves smoothly (e.g., temporal shifts or spatial differences). In such settings, discrete reweighting often produces unstable training dynamics or suboptimal performance due to noisy or imbalanced domain counts. Our continuous method offers a principled alternative that yields more robust gradient estimates, better utilization of cross-domain information, and improved generalization performance, as confirmed by our experimental results.
>
> We have added this discussion to the Appendix in the revised version as suggested
>
> [1] Xu et al. Domain-Indexing Variational Bayes: Interpretable Domain Index for Domain Adaptation. ICLR 2023

---

### Official Review · Reviewer_hTEs · 2025-10-31

**Soundness:** 2
**Presentation:** 2
**Contribution:** 3
**Rating:** 2
**Confidence:** 3

**Summary:**

This paper proposes a realistic online multi-domain adaptation setting where each training round receives only a subset (k ≪ C) of domains with domain imbalance. It extends VDI by (1) introducing a time-varying prior to encourage temporal continuity, and (2) designing Domain-Index-Aware Reweighting to mitigate data imbalance. The method is evaluated on Growing Circle, CompCars, and TPT-48. On all of these datasets, the proposed method beats all baselines.

**Strengths:**

* This is a realistic problem for online products: partial access + imbalance + drift.
* The design is simple and easy to extend VDI.
* The gains in both in-round and next-round evaluation are consistent.

**Weaknesses:**

* Major: Motivation is online indexing for ads/recs, but no CTR/CVR datasets are used. Circles/cars/temperature don’t guarantee benefits for CTR/CVR datasets.

* No ablation studies about the importance of each component: temporal prior vs. DIAR vs. replay.

* The paper mentions dynamic and resource-constrained conditions in the abstract but doesn’t discuss this in the results.

* Most shifts are smooth; it should be tested on harsher cases such as abrupt shifts, tiny k, and extreme imbalance.

* TPT-48 tables label MSE as “Accuracy.”

**Questions:**

Please check Weaknesses.

---

> ### Author Response · Authors · 2025-11-21
> **[1/2] Rebuttal**
>
> Thank you for your constructive comments. We are glad that you find our problem `"realistic"`, our design `"simple and easy"`, and the performance improvement `"consistent"`. Below, we address your questions one by one in detail. We have also **included all discussions below in our revision** (with the changed part marked in blue).
>
> >**Q1: "Major: Motivation is online indexing for ads/recs, but no CTR/CVR datasets are used. Circles/cars/temperature don’t guarantee benefits for CTR/CVR datasets."**
>
> We apologize for the confusion. The ads and recommendation examples in the paper are meant only as motivating scenarios. Our contribution is a fundamental and general online domain adaptation method under domain imbalance, and therefore is not limited to CTR/CVR.
>
> As an alternative, below we provide another motivating example, which can directly replace the ads/recs examples in our abstract if preferred:
>
> For example, climate or environmental sensor networks deployed across widely separated regions form multiple domains whose data do not arrive synchronously. Nearby regions may provide timely data, while distant regions (e.g., collecting African climate signals from a U.S.-based server) may exhibit substantial delays. Thus, at each time step, only a partially observed and highly imbalanced subset of domains is available (e.g., 2–3 out of 10), while all domains evolve over time due to environmental changes. This multi-domain, time-varying, partially observable structure aligns naturally with our ODI formulation and goes beyond the assumptions of traditional domain adaptation.
>
> We have added this clarification and the alternative example to the revised version of the paper.
>
> >**Q3: "The paper mentions dynamic and resource-constrained conditions in the abstract but doesn’t discuss this in the results."**
>
> We are sorry for the confusion. We would like to clarify that our problem setting already incorporates both dynamic and resource-constrained conditions:
>
> **(1) Dynamic setting.** Our ODI setting is inherently dynamic in three ways:
>
> - *Dynamic distributions:* At each time step $t$, all domains $d_1, d_2, ..., d_{10}$ evolve over time (e.g., across $20$ time steps). Thus each domain has a temporally shifting distribution.
> - *Dynamic partial observability:* Although 10 domains exist, only a random subset (e.g., 3–5 domains) is observed at each time step, making the domain stream partially observable.
> - *Dynamic imbalance:* Even among the observed domains, the available samples are extremely imbalanced, further increasing the difficulty of adaptation.
>
> These elements together create a multi-domain, time-evolving, partially observed, imbalanced environment that is much more dynamic than classical DA settings.
>
> **(2) Resource-constrained setting.** If resources were unlimited, one could simply aggregate all historical data and retrain the model from scratch at every time step. This would largely mitigate imbalance and instability.
> However, in realistic scenarios, as the number of time steps grows, continuously scaling the training set is computationally prohibitive, making full retraining infeasible. This is precisely why we introduce the domain-aware replay buffer:
> we selectively retain a subset of past samples and perform online adaptation on top of them, enabling effective domain adaptation under strict resource limitations.
>
> We have added this clarification to the revised version of the paper.
>
> >**Q4: "Most shifts are smooth; it should be tested on harsher cases such as abrupt shifts, tiny k, and extreme imbalance.”"**
>
> Thank you for mentioning this. We would like to clarify that the assumption that most shifts are smooth is standard in the literature, including VDI, CIDA, and the CDA paper referenced by Reviewer 21zP [1,2,3].
>
> Note that our datasets indeed contain harsh distribution shifts that you mentioned. For example, in CompCars we treat the front-view car images as the source domain, while other viewpoints (e.g., rear view, side view) serve as target domains. These cross-view domain transitions are highly non-smooth and introduce substantial distribution gaps.
>
> We will add the discussions above in the Appendix of the revision as suggested
>
> >**Q5: "TPT-48 tables label MSE as “Accuracy.”"**
>
> We are sorry for the typo, and have fixed them in the revision.
>
> [1] Domain-Indexing Variational Bayes: Interpretable Domain Index for Domain Adaptation. ICLR 2023
>
> [2] Continuously Indexed Domain Adaptation. ICML 2020
>
> [3] Delving into the Continuous Domain Adaptation, MM, 2022

---

> ### Author Response · Authors · 2025-11-22
> **[2/2] Rebuttal**
>
> >**Q2: "No ablation studies about the importance of each component: temporal prior vs. DIAR vs. replay."**
>
> This is a good point. Following your suggestion, we are running additional ablation studies for (1) temporal priors **(P)**, (2) domain-index-aware reweighting **(R)**, and (3) replay buffer **(B)** on the CompCar dataset with 5 time steps (i.e., 5 temporal rounds), and Table A & B below summarizes our new results.
>
> Table A. Ablation Studies. In-round accuracy on CompCars (with 5 temporal rounds) comparing our full ODI, VDI, and ODI without temporal priors **(P)**, domain-index-aware reweighting **(R)**, or replay buffer **(B)**.
>
> | Model | 1 | 2 | 3 | 4 | 5 | Average |
> |------------------|---------:|-------:|-------:|------:|--------:|---------:|
> | **VDI-O** | 46.0 | 39.5 | 34.8 |  41.6 | 34.1 |  39.2  |
> | **ODI w/o P** | 46.7 | 44.1 | 42.8 |  41.4 | 41.3 | 43.3 |
> | **ODI w/o B** | 45.6 | 39.4 | 37.4 | 41.8 | 35.4 | 40.0 |
> | **ODI w/o R** | 45.0 | 44.3 | 35.7 | **42.6** | 34.4 | 40.4 |
> | **ODI** | **48.4** | **45.2** | **44.0** | 41.9 | **41.9** | **44.3** |
>
> Table B. Ablation Studies. Next-round accuracy on CompCars comparing our full ODI, VDI, and ODI without temporal priors **(P)**, domain-index-aware reweighting **(R)**, or replay buffer **(B)**.
>
> | Model | 1 | 2 | 3 | 4 | 5 | Average |
> |------------------|---------:|-------:|-------:|------:|--------:|---------:|
> | **VDI-O** | **45.7** | 40.0 | 35.6 |  33.0 | 38.6 | 38.6 |
> | **ODI w/o P** | 44.1 | 41.6 | 40.7 |  38.2 | 39.5 | 40.8 |
> | **ODI w/o B** | 45.3 | 40.8 | 40.2 | 36.9 | 38.8 | 40.4 |
> | **ODI w/o R** | **45.7** | 40.4 | 41.1 | 33.7 | 39.9 | 40.2 |
> | **ODI** | **45.7** | **43.9** | **41.2** | **40.0** | **40.6** | **42.1** |
>
> As shown in Tables A and B, the full ODI model consistently outperforms all its ablated variants across both in-round and next-round evaluations. These results highlight the complementary roles of all three components:
>
> - **Temporal prior (P)** significantly improves stability and carries useful information forward across rounds. Removing it (ODI w/o P) leads to noticeable drops in both current- and next-round accuracy.
> - **Domain-index-aware reweighting (R)** is essential for handling domain-wise imbalance. Without it (ODI w/o R), the model becomes overly biased toward the majority domains and fails to generalize across rounds.
> - **Replay buffer (B)** is particularly important under partial observability. Removing it (ODI w/o B) harms the model’s ability to retain information about previously unseen or lightly sampled domains.
>
> Importantly, ODI outperforms VDI-O by a clear margin, demonstrating that simply applying an offline DA method (VDI-O) to the online setting is insufficient. All three online components, i.e., temporal priors, domain-index-aware reweighting, and replay buffering, are necessary to handle distributional shift, partial observability, and domain imbalance in online environments.
>
> We will include these additional results and analyses in the Appendix of the revised paper.

---

### Official Review · Reviewer_9rYZ · 2025-11-01

**Soundness:** 3
**Presentation:** 3
**Contribution:** 3
**Rating:** 6
**Confidence:** 3

**Summary:**

This paper introduces a new setting, online domain adaptation learning with imbalanced labels. They proposed a new framework with (1) temporal priors over domain indices, and (2) domain-index-aware reweighting to tackle the challenge. Abundant experiments are conducted to validate the method's effectiveness. Overall, this paper is interesting and technically sound.

**Strengths:**

1. Easy to read, easy to follow, well organized.
2. The new problem formulation is well defined and close to the real application.
3. The framework design is technically sound.

**Weaknesses:**

1. Please add more details about the experiment implementation. More specifically, how do you simulate "online scenarios with distributional shift, partial observability, and data sparsity"?
2. How about the method's performance in the traditional DA setting? I think it is fair to compare ODI and VDI in the previous DA setting.
3. Ablation study is missed. Please analyze the contribution of (1)  temporal priors, (2) domain-index-aware reweighting, and (3) replay buffer?

**Questions:**

Please answer the question in Weakness.

---

> ### Author Response · Authors · 2025-11-21
> **[1/2] Rebuttal**
>
> Thank you for your constructive comments. We are glad that you find our paper `"Easy to read, easy to follow, well organized"`, our problem formulation `"well defined and close to the real application"`, and our framework design `"technically sound"`. Below, we address your questions one by one in detail. We have also **included all discussions below in our revision** (with the changed part marked in blue).
>
> >**Q1: "Please add more details about the experiment implementation. More specifically, how do you simulate "online scenarios with distributional shift, partial observability, and data sparsity"?"**
>
> Thank you for mentioning this. We clarify the implementation details of our online setting below:
>
> **Distributional Shift (Online Evolution Across Time).** We simulate an online environment with $T$ rounds and $K$ domains. As time progresses, each domain’s distribution evolves, instead of remaining fixed as in traditional DA settings (which typically assume a single source and a single target). This produces **continuous temporal drift** within each domain, naturally inducing an online distributional shift. Note that in our setting such shift happens for all $K$ domains, as shown in Figure 1 of the paper.
>
> **Partial Observability (Only a Subset of Domains Available at Each Time).** Although $K$ domains exist globally, in realistic scenarios not all domains provide data at every round. Therefore, at each round $t$, we randomly sample **only a small subset** of domains to be observed (e.g., 2–3 out of 10), while the remaining domains contribute no data. This mimics real-world situations where data from some domains arrive with delay or are temporarily missing, resulting in partial observability.
>
> **Data Sparsity and Imbalanced Access.** Even among the domains observed at round $t$, the number of available samples is highly imbalanced. While an ideal scenario might yield a balanced allocation (e.g., 100 samples per domain), in practice some domains may contribute 80 samples, others only 10 or even fewer. This creates severe data sparsity and domain-level imbalance, which ODI is specifically designed to address.
>
> >**Q2: "How about the method's performance in the traditional DA setting? I think it is fair to compare ODI and VDI in the previous DA setting."**
>
> We are sorry for the confusion. We would like to clarify the key differences between traditional domain adaptation (DA) and our Online Domain Index (ODI) setting:
>
> + In the traditional DA setting, one usually assumes $K$ continuous domains. This is a **one-dimensional** domain structure.
> + In our ODI setting, we assume $K$ continous domains **for each time step $t\in \{1,\dots, T\}$**. This is a **two-dimensional** domain structure. For example, with 10 domains across 20 time steps, the data consists of total 200 domains.
>
> | t | $D_1$ | $D_2$ | $D_3$ | ... | $D_{10}$ |
> |---|-----|-----|-----|-----|-----|
> |1| $D_{1}^{1}$ | $D_{2}^{1}$ | $D_{3}^{1}$ | ... | $D_{10}^{1}$ |
> |2| $D_{1}^{2}$ | $D_{2}^{2}$ | $D_{3}^{2}$ | ... | $D_{10}^{2}$ |
> |3| $D_{1}^{3}$ | $D_{2}^{3}$ | $D_{3}^{3}$ | ... | $D_{10}^{3}$ |
> |...| ... | ... | ... | ... |... |
> |20| $D_{1}^{20}$ | $D_{2}^{20}$ | $D_{3}^{20}$ | ... | $D_{10}^{20}$ |
>
> Each domain evolves over time, and different domains also evolve differently. Thus, ODI involves **jointly time-varying and domain-varying distributions**.
>
> Our ODI method is specifically designed for this much more complex, multi-domain, time-evolving scenario, and therefore is **not applicable** to the traditional DA setting. Note that our **main baseline VDI** is **not applicable** to the traditional DA setting **either**.
>
> We will added the disucssion above into the Appendix of the revision.

---

> > ### Comment · Reviewer_9rYZ · 2025-11-26
> >
> > Thank you for the rebuttal. Most of my concerns have now been addressed, so I intend to maintain my original rating

---

> > > ### Author Response · Authors · 2025-11-26
> > > **Thank you**
> > >
> > > Dear Reviewer 9rYZ,
> > >
> > > Thank you once again for your encouraging and valuable feedback! We are glad that our response has addressed your concerns. We will be sure to incorporate your comments and our associated response in the final revision.
> > >
> > > Best regards,
> > >
> > > ODI Authors

---

> ### Author Response · Authors · 2025-11-22
> **[2/2] Rebuttal**
>
> >**Q3: "Ablation study is missed. Please analyze the contribution of (1) temporal priors, (2) domain-index-aware reweighting, and (3) replay buffer?"**
>
> This is a good point. Following your suggestion, we are running additional ablation studies for (1) temporal priors **(P)**, (2) domain-index-aware reweighting **(R)**, and (3) replay buffer **(B)** on the CompCar dataset with 5 time steps (i.e., 5 temporal rounds), and Table A & B below summarizes our new results.
>
> Table A. Ablation Studies. In-round accuracy on CompCars (with 5 temporal rounds) comparing our full ODI, VDI, and ODI without temporal priors **(P)**, domain-index-aware reweighting **(R)**, or replay buffer **(B)**.
>
> | Model | 1 | 2 | 3 | 4 | 5 | Average |
> |------------------|---------:|-------:|-------:|------:|--------:|---------:|
> | **VDI-O** | 46.0 | 39.5 | 34.8 |  41.6 | 34.1 |  39.2  |
> | **ODI w/o P** | 46.7 | 44.1 | 42.8 |  41.4 | 41.3 | 43.3 |
> | **ODI w/o B** | 45.6 | 39.4 | 37.4 | 41.8 | 35.4 | 40.0 |
> | **ODI w/o R** | 45.0 | 44.3 | 35.7 | **42.6** | 34.4 | 40.4 |
> | **ODI** | **48.4** | **45.2** | **44.0** | 41.9 | **41.9** | **44.3** |
>
> Table B. Ablation Studies. Next-round accuracy on CompCars comparing our full ODI, VDI, and ODI without temporal priors **(P)**, domain-index-aware reweighting **(R)**, or replay buffer **(B)**.
>
> | Model | 1 | 2 | 3 | 4 | 5 | Average |
> |------------------|---------:|-------:|-------:|------:|--------:|---------:|
> | **VDI-O** | **45.7** | 40.0 | 35.6 |  33.0 | 38.6 | 38.6 |
> | **ODI w/o P** | 44.1 | 41.6 | 40.7 |  38.2 | 39.5 | 40.8 |
> | **ODI w/o B** | 45.3 | 40.8 | 40.2 | 36.9 | 38.8 | 40.4 |
> | **ODI w/o R** | **45.7** | 40.4 | 41.1 | 33.7 | 39.9 | 40.2 |
> | **ODI** | **45.7** | **43.9** | **41.2** | **40.0** | **40.6** | **42.1** |
>
> As shown in Tables A and B, the full ODI model consistently outperforms all its ablated variants across both in-round and next-round evaluations. These results highlight the complementary roles of all three components:
>
> - **Temporal prior (P)** significantly improves stability and carries useful information forward across rounds. Removing it (ODI w/o P) leads to noticeable drops in both current- and next-round accuracy.
> - **Domain-index-aware reweighting (R)** is essential for handling domain-wise imbalance. Without it (ODI w/o R), the model becomes overly biased toward the majority domains and fails to generalize across rounds.
> - **Replay buffer (B)** is particularly important under partial observability. Removing it (ODI w/o B) harms the model’s ability to retain information about previously unseen or lightly sampled domains.
>
> Importantly, ODI outperforms VDI-O by a clear margin, demonstrating that simply applying an offline DA method (VDI-O) to the online setting is insufficient. All three online components, i.e., temporal priors, domain-index-aware reweighting, and replay buffering, are necessary to handle distributional shift, partial observability, and domain imbalance in online environments.
>
> We will include these additional results and analyses in the Appendix of the revised paper.

---

### Author Response · Authors · 2025-11-25
**Response to All Reviewers**

Dear reviewers,

We would like to thank all reviewers for their efforts in reviewing our paper and providing valuable, constructive comments. We are glad that the reviewers find that

+ our paper `"easy to read, easy to follow, well organized"` (9rYZ)
+ our problem formulation `"well defined and close to the real application"` (9rYZ) / `"sound and from the real-world application scenarios"` (21zP) / `"realistic"` (hTEs)
+ our framework design `"technically sound"` (9rYZ) / `"simple and easy"` (hTEs)
+ our optimization and lower bound `"correct"` (21zP)
+ the performance improvement `"consistent"` (hTEs)

We have also tried our best to address the reviewers' questions one by one in detail. Below, we summarze our responses to the main concerns of reviewers.

+ Reviewer 9rYZ: We **added ablation studies** to demonstrate the contribution of each component, **clarified the experimental implementations and problem settings**, and **highlighted the differences** between our setting and traditional Domain Adaptation settings.
+ Reviewer 21zP: We **added another baseline (i.e., CDA)** and corresponding citations, **specified that our setting is realistic**, **highlighted the key aspects of our problem formulation**, **clarified the scale of our experiments**, and **demonstrated that our continuous domain-reweighting technique is non-trivial**.
+ Reviewer hTEs: We **added ablation studies** to show the contribution of each component, **clarified the motivation** behind our method, **outlined the assumptions and constraints** in our problem setting, and **corrected typos**.

We have also incorporated these additional experiments and discussions into the revision.

We hope these clarifications fully address the reviewers’ concerns, and we will be more than happy to answer any of the reviewers' follow-up questions, if any.

Last but not least, we would also like to to sincerely thank the AC and SAC for their time and effort throughout the review process.

---

### Meta-Review · Area_Chair_oePj · 2025-12-13

**Summary:**

This paper proposes Online Domain Indexing (ODI) for online domain adaptation under partial domain observability and severe inter domain imbalance, extending the VDI style domain indexing idea with (i) a temporal prior over indices, (ii) domain index aware reweighting, and (iii) a domain aware replay buffer. The initial reviews are mixed but leaning negative: one reviewer is positive at 6 (9rYZ), one is borderline at 4 (21zP), and one is negative at 2 (hTEs).

The rebuttal meaningfully improves the paper. It adds concrete details on how the online protocol is simulated, adds ablations for the three main components, fixes minor issues, and implements an adapted CDA baseline requested by 21zP. However, even after the rebuttal, the main weakness is that the core technical novelty feels incremental relative to prior domain indexing and continual or continuous DA threads. The setting is more complex, but the method mostly combines expected ingredients (temporal smoothness, reweighting, replay) and the empirical study still does not fully settle whether ODI brings a clearly new principle versus a solid engineering extension. Given the current score distribution and remaining novelty concerns, my decision is reject.

**Reviewer Concerns:**

Concerns that have been addressed well:

1. Missing implementation details and protocol clarity (9rYZ): The rebuttal gives a clear description of temporal drift, partial observability, and imbalance simulation, plus adds requested ablations.

2. Ablations (9rYZ, hTEs): Added and the trends are consistent. Each component appears to contribute, especially the temporal prior and the reweighting.

3. Missing baseline line of work (21zP): They cite the continuous DA paper and implement an adapted CDA baseline, with ODI still ahead in the provided tables.

4. Minor presentation issues (hTEs): Typos and table labeling were fixed.

Concerns that remain outstanding:

1. Novelty relative to prior art (21zP): Even with clarifications on continuous domain reweighting, the method still reads as an extension of VDI plus replay and weighting. The paper does not convincingly show a new idea that would generalize beyond this specific construction.

2. Strength of evidence under harsher regimes (hTEs): The rebuttal argues CompCars already includes non smooth shifts, but the paper still lacks a clean stress test suite where k is very small, imbalance is extreme, or shift is abrupt, reported in a way that clearly changes the overall conclusion.

**Reviewer Scores:**

Reviewer 9rYZ: stays at 6 (they explicitly said they will maintain their rating after rebuttal).

Reviewer 21zP: likely stays around 4, as their main point on limited novelty may remain.

Reviewer hTEs: likely stays at 2, since their main complaint is about motivation framing and they also asked for harsher tests.

Overall, the paper does not reach a clear acceptance level across reviewers, and the novelty concern remains the key blocker. My recommendation is reject.

---

### Decision · Program_Chairs · 2026-01-26

Reject